# Designing Lightweight Stadium Roofing Structures Based on Advanced Analysis Methods

**Faham Tahmasebinia *** [ID]**, Eason Chen** [ID]**, Andy Huang and Jesse Li**

School of Civil Engineering, The University of Sydney, Sydney, NSW 2006, Australia
* Correspondence: faham.tahmasebinia@sydney.edu.au

**Abstract:** The current structural engineering practical standards are unable to offer an universal structural design standard for long-spanning lightweight stadium roofing structures. As such, the design procedure of a particular stadium roof is not replicable to another. This research aims to present a novel design procedure for lightweight stadium roofing structures considering the Lakhwiya stadium the Optus Stadium and the CommBank Stadium as experimental cases. Using the finite element analysis (FEA) software Strand7, the cases will be modelled and analysed. Varying load cases and combinations such as ultimate strength (ULS) and serviceability limit states (SLS) based on the Australian Standard AS1170.0:2002 will be calculated and subsequently applied. Linear static analysis will then be undertaken where critical members will be identified within the model. Based on this, preliminary member sizing and design feasibility checks will be conducted in order to ensure structural stability and compliance to the Australian Steel Structure code AS4100:2020. A linear buckling analysis is also conducted based on the selected sizes from the initial stage to determine critical loads. Advanced analysis including non-linear buckling computation is comprehensively managed. Some of the crucial parameters such as maximum displacement, maximum/minimum principal stresses, critical buckling loads, as well as load factors are examined. The main novelty of this study is to determine a clear road map to design stadium roofing systems subjected to a combination of different types of the loading.

**Keywords:** stadium roof; finite element analysis; lightweight structure

## 1. Introduction

### 1.1. Problem Specification

The number of modern stadium roofs that have opted for longer-spanning architectural designs has significantly increased over the past few decades. This may be attributed to factors such as increases in population density, increased demand in sporting events, enhanced aesthetics, and wider socio-economic trends [1,2]. In addition, lightweight and reusable materials are preferably utilised in infrastructure construction [3]. Lightweight materials provide the advantages of having reduced costs, increased design flexibility, and decreased construction time [4]. However, as modern stadium roofing structures are becoming progressively longer in span, structural issues such as strength and serviceability issues will arise. This is due to them having minimal supporting structures for the roof to maximise visibility and aesthetics [5]. In addition, as the members selected for stadium roofing structures are lightweight, thin-walled members, buckling is also prone to occur prior to member yielding [6]. As such, it is widely known that the designing of long-span lightweight stadium roofing structures will result in structural difficulties, and corollary, there is no generalised design procedure for designing stadium roofs in the industry. Civil engineers are often forced to rely on instinct and previous experience during the design cycle, and the design procedure of a particular stadium is often not replicable to another.

*1.2. Objectives*

Since the design process for stadium roofs needs to be better established, this paper aims to provide a general guideline for the design procedure of stadium roofs. This can be achieved by designing three models of a stadium roofing structure, namely the CommBank, Optus, and Lakhwiya stadium [1], based on the process in Strand7 as an example case. Linear and nonlinear analyses are performed in Strand7 to evaluate the capacity and feasibility of using predetermined member sizes. This paper proposes suggestions for the future research direction at the end of the project. The detailed objectives are presented as the following:

- Design a three-dimensional structural model of a stadium roof based on an existing stadium.
- Select a lightweight material and assign initial member sizes for structural members.
- Set up load combinations for FEM analysis (i.e., dead loads, live loads, and wind loads) according to Australian Standards.
- Perform linear static, linear buckling, and nonlinear static analyses of the model to gain maximum deflections and critical loads of the model.
- Verify the capacity of the critical member according to Australian Standards.

*1.3. Scope*

This report will consist of four major sections: materials and methodology of finite element analysis, linear static, linear buckling, and non-linear buckling results, and a holistic discussion. During the linear static analysis, varying combinations of dead, live, and wind load will be applied during all stages of the analysis based on the Australian Standards for load combinations, AS1170.2:2002 [7]. Design feasibility checks of the pre-selected section sizes will be conducted based on AS4100:2018 [8]. This will be conducted in combination with the output critical members of the top, and bottom Chord, diagonals, and cross-bracing based on the axial forces of tension and compression during the initial linear static analysis. In addition, the selected members will also be evaluated under flexural conditions. Linear buckling analysis will then be conducted based on each load combination under linear static analysis and the critical buckling load will be determined. Non-linear buckling analysis will then be undertaken based on the first mode shape in linear buckling analysis and comparisons will be made based on parameters such as global displacement, critical buckling load, and critical members. Based on this, a universal design procedure will be presented in the form of a flowchart.

## 2. Materials and Methods

*2.1. Materials*

Cold-formed stainless steel such as Litesteel shows good resistance against corrosion and requires less maintenance cost during its service life [9–11]. To design lightweight stadium roofing structures consisting of trusses, square hollow sections (SHS) and circular hollow sections (CHS) will be examined. Cold-formed square hollow section (SHS) is formed by cold-rolling with a weld of the annealed flat strip into a circular hollow section and then further rolled into SHS [12,13]. Due to the characteristics of hollow sections having a lower mass per meter resulting in reduced self-weight whilst exhibiting excellent structural properties under tension and compression when assembled as a truss, they are often utilised in lightweight, long-spanning roofing structures [14]. As the structure consists of a truss, hollow sections will be preferred as members within a truss experience little to no bending, and tensile or compressive forces often govern design. Hollow sections perform well under tension and compression; however, they are inferior under bending compared with the I-sections [15].

The use of hollow sections within a truss section will also result in increased architectural aesthetics as hollow sections are often assembled using lattice construction where the members are directly connected to each other in the absence of other elements such as connecting plates. In addition, due to the longer spans of stadium roofing structures, the use of SHS compared with I-beams will result in slender and reduced section depths and

further increasing the aesthetic appeal and decreasing overall structural self-weight [15]. Therefore, SHS were selected during the modelling process and its material properties and preliminary section sizes are shown below in Table 1. Note that the initial section of member sizes and the structures of stadium models are based on the estimation rather than the exact data, and relevant structures designs could be refereed to Section 2.3 of this paper.

**Table 1.** Preliminary Section Sizes.

| Stadium | Frame | Structural Member | Member Size |
|---|---|---|---|
| CommBank Stadium (SHS) | Main | Top Chord | C450L0 150 × 150 × 6 |
| | | Bot Chord | C450L0 150 × 150 × 8 |
| | | Diagonal | C450L0 150 × 150 × 5 |
| | Connection | Top Chord | C450L0 125 × 125 × 5 |
| | | Bot Chord | C450L0 125 × 125 × 6 |
| | | Diagonal Chord | C450L0 125 × 125 × 4 |
| Optus Stadium | Main | Top Chord | C350L0 273.1 × 12.7 |
| | | Bot Chord | C350L0 273.1 × 12.7 |
| | | Web Diagonal | C350L0 101.6 × 3.2 |
| | | Bot Diagonal | C350L0 101.6 × 3.2 |
| | | Bot Structure | C350L0 168.3 × 7.1 |
| | Connection | Lateral Beam | C350L0 88.9 × 2.6 |
| Lakhwiya Stadium | End Frame | Top Chord | C450L0 200 × 200 × 8 |
| | | Bot Chord | C450L0 200 × 200 × 9 |
| | | Diagonals | C450L0 65 × 65 × 6 |
| | | Cross-Bracing | C450L0 65 × 65 × 3 |
| | Central Frame | Top Chord | C450L0 200 × 200 × 8 |
| | | Bot Chord | C450L0 200 × 200 × |
| | | Diagonals | C450L0 100 × 100 × 4 |
| | | Cross-Bracing | C450L0 65 × 65 × 3 |

*2.2. Finite Element Analysis*

2.2.1. Defining Load Combinations

According to AS1170.0:2002 [16], a structure must satisfy load combination cases in the context of ultimate limit state (ULS) and also serviceability limit state (SLS). A total of three load combinations including permanent and imposed action, permanent action only and permanent, wind reversal, and imposed actions will be considered for ULS [17]. The load combination of permanent, short-term, and long term imposed in addition to wind will be considered for SLS. Table 2 below exhibits all load cases under consideration.

**Table 2.** Load Combinations under Consideration.

| Limit State | Load Combination |
|---|---|
| Ultimate Limit State (ULS) | $1.2G + 1.5Q$ <br> $1.35G$ <br> $1.2G + W + \Psi_c \, ^1Q$ |
| Serviceability Limit State (SLS) | $G + \Psi_s \, ^2Q + Wup$ <br> $G + \Psi_l \, ^3Q + Wup$ |

[1] $\Psi_c$ is taken as 0 as stadium roofs are classified as "All other roofs" in accordance with AS1170 Table 4.1.; [2] $\Psi_s$ is the short-term factor and is taken as 0.7; [3] $\Psi_l$ is the long-term factor and is taken as 0.

2.2.2. Linear Analysis

Upon completion of the construction of the modelled stadiums, the load cases as defined above in Section 2.2.1 were applied according to AS1170.0:2002 [16]. Based on these applied loads, linear static analysis was firstly conducted using the solver function build in Strand7 [18]. Under linear static analysis, the solver assumes the structure will remain within the linear elastic range. This implies that the truss members are able to return to their original shape when the applied force is absented. As such, the analysis assumes a linear relationship between the applied forces and associated outputs such as displacement, axial

forces, and bending. Using the outputs of the linear static analysis, critical members under tension, compression and flexural forces under critical load cases will be identified and the design feasibility of preliminary member sizes will be verified based on the Australian Steel Structures code AS4100:2020 [8].

2.2.3. Linear Buckling vs. Nonlinear Buckling Analysis

This section of the report will provide a theoretical overview and description of the contrasting nature between linear and non-linear buckling analysis. As the stadium roof structure utilizes slender, lightweight, thin-walled members of cold-formed square hollow section and circular hollow sections, buckling is prone to occur prior to yield [6]. In addition, buckling failure is characterised as being sudden with minimal prior warning which is undesirable in the context of structural design as this poses a significant threat to human life [19]. Therefore, it is essential to conduct buckling analysis to gain an understanding of the behaviour of the structure.

Linear Buckling

Under linear buckling analysis, the structure's material is assumed to remain in the elastic region of the stress–strain curve, correlating to a linear relationship between stress and strain [20]. In addition, it is also assumed that the stresses within the structure increase in proportion with the applied load with relatively minimal deformation or geometrical change. Under this scenario, a bifurcation point exists where primary linear load paths and secondary post buckling load paths intersect (Strand7) [21]. This point is the intersection between the pre-buckling of the structure and the post buckling of the structure. It should be noted that the secondary path or the post-buckling path no longer exhibits a linear relationship between applied load [22] and displacement, as will be discussed below in Section 3.3. Until the bifurcation point, the structure conforms to the Euler column formula where the critical buckling load is dependent upon parameters such as the elastic modulus, second moment of area, member length and effective length. The linear buckling solver within Strand7 is based on the results from a single load-case of the linear static solver and attempts to calculate the critical buckling load by solving the mathematical solution to the eigenvalue of the stiffness matrix defined below in Equation (1) [23].

$$[(K) - \lambda(K_g)]\{\delta\} \tag{1}$$

where K: global elastic stiffness matrix; $K_g$: geometric stiffness matrix; $\lambda$: eigenvalue; $\delta$: eigenvector.

The obtained eigenvalue through linear buckling analysis denotes the critical buckling load-factor where the critical buckling load is determined by multiplying the applied load with the obtained eigenvalue [24]. This can be performed simply as the applied forces have been predetermined (Strand7) [21].

Non-Linear Buckling

Prior to reaching the bifurcation point, or during the pre-buckling phase the structure is assumed to exhibit linear behaviour. However, post-buckling behaviour of a structure in reality is non-linear and is dependent on factors including geometric and physical nonlinearities [25–28]. This was considered in Strand7's nonlinear static solver through the selection of geometric nonlinearity "GNL" and material nonlinearity "MNL" during the non-linear buckling analysis.

In addition, as the structure enters post-buckling it exhibits inelastic behaviour meaning a stress–strain curve must be defined within the solver. The stress–strain curve is shown below in Figure 1 for all three stadiums.

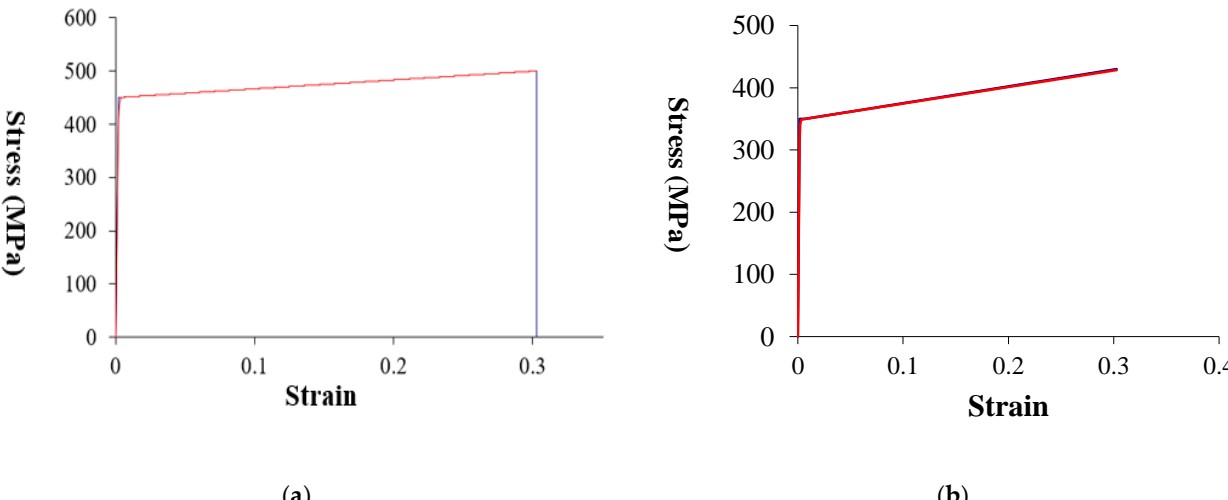

**Figure 1.** Stress–strain curve for nonlinear analysis: (**a**) stress–strain Curve for Lakhwiya and CommBank Stadium (C450L0); (**b**) stress–strain Curve for Optus Stadium (C350L0).

Using the defined stress–strain curve and considerations for material and geometric nonlinearity, the non-linear static solver was run using the deformed shape of "mode 1" from the linear buckling solver. Contrary to the linear solver, the non-linear static solver will apply the loads via pre-determined loading increments and load factors as to simulate the nonlinear buckling behaviour. Therefore, loading factors and increments must be pre-determined prior to running the non-linear solver, [29,30]. For each load combination, the maximum loading factor for the initial non-linear analysis was entered as the eigenvalue obtained from the linear buckling solver, and increments were defined from 0 to the maximum loading factor with equal increases of increments [31]. The eigenvalue from the linear analysis was used as the maximum loading factor during the initial stages of the analysis as the calculated critical buckling load under linear assumptions are often larger in magnitude and overestimates the critical buckling load in comparison to non-linear analysis and therefore results in fewer conservative results [32]. Therefore, it will suffice to use the eigenvalue as the maximum loading factor within the non-linear solver for the initial analysis. The maximum loading factor and increments during the later analyses were then adjusted accordingly based on the initial analysis.

The solver will then attempt to solve Equation (2) shown below, a linear equation system for (Δd) for each pre-defined increments up until the maximum loading factor which is a method based on the modified Newton–Raphson method (Strand7) [33,34].

$$[K(d, \sigma_e, \varepsilon_e)(\Delta_d)] = R \tag{2}$$

where $K (d, \sigma_e, \varepsilon_e)$: current global stiffness matrix; $\Delta_d$: displacement increment vector; $R = (P) - F (d, \sigma_e, \varepsilon_e)$; where: R: global residual force vector or unbalanced vector; F: $(d, \sigma_e, \varepsilon_e)$; (P): current external force vector

Following this, convergence is then checked based on Equations (3) and (4) below which correlates to displacement norm and residual force norm, respectively (Strand7) [35].

$$\|\Delta d\| / \|d\| < \varepsilon_e \tag{3}$$

$$\|R\| / \|P0\| < \varepsilon_r \tag{4}$$

where $\|\Delta d\|$: is the norm of increment displacement vector; $\|d\|$: is the norm of total displacement vector; $\|P0\|$: is the norm of the residual force vector at the first iteration; $\|R\|$: is the norm of the residual force vector at the current iteration; $\varepsilon_e$: is the convergence tolerance on displacement force; $\varepsilon_r$: is the convergence tolerance on residual force.

When Equations (3) and (4) are satisfied, the non-linear solver will then commence solving for the next sub increment. This will be repeated until the maximum loading factor has been applied. The onset of buckling within the non-linear static solver can be determined through visual inspection of the convergence graph where the solver will struggle to converge at a given increment. In addition, the critical buckling loading factor can also be determined by creating a graph of loading factor vs. nodal displacement, where a change in gradient of the graph will indicate buckling. Using the loading factor, the critical non-linear buckling load can then be calculated by multiplying the factor by the pre-determined load cases.

### 2.3. Models

Figures 2–7 below illustrate the 2D Planar view of a single truss member and 3D holistic isometric views of the modeled stadiums within Strand7 [36–38].

The design process of the CommBank Stadium Roof refers to existing structure and the whole progress was performed via using Strand7. The first step is to create one single truss element in 2D plane (Figure 2) and convert to 3D plan with global coordinates and copy by 13 increments (i.e., 14 Trusses in total for one side) to form one side of the structure. The whole structure is similar to a rectangle with angular corner. Corner sections could be connected by creating a cylindrical coordinate system and creating copy by increments with specific angles. In this model, the trusses span 35 m inward and 10 m outward of the stadium with 5 m of maximum height. There are 14 trusses along the long side and 6 trusses along the short side and each corner consists of 4 trusses which means the model has 54 trusses or 53 bays in total.

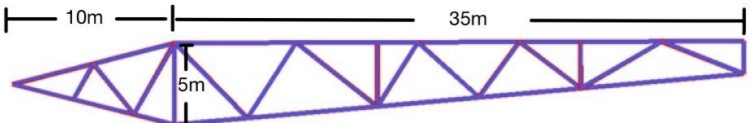

**Figure 2.** Two-Dimensional View of Singular Truss Member of CommBank Stadium.

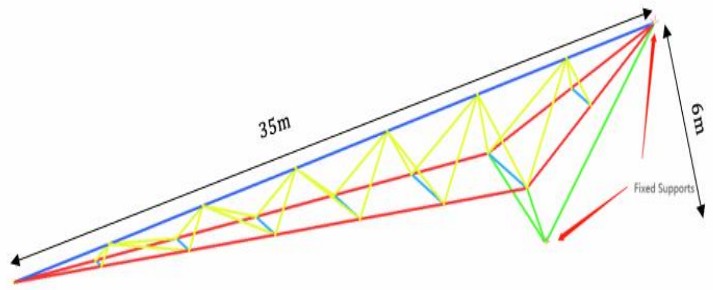

**Figure 3.** Two-Dimensional View of Singular Truss Member of Optus Stadium.

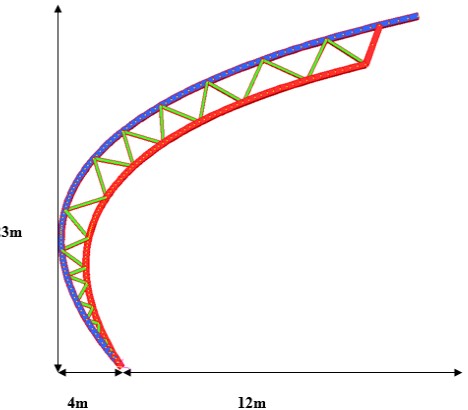

**Figure 4.** Two-Dimensional View of Singular Truss Member of Lakhwiya Stadium.

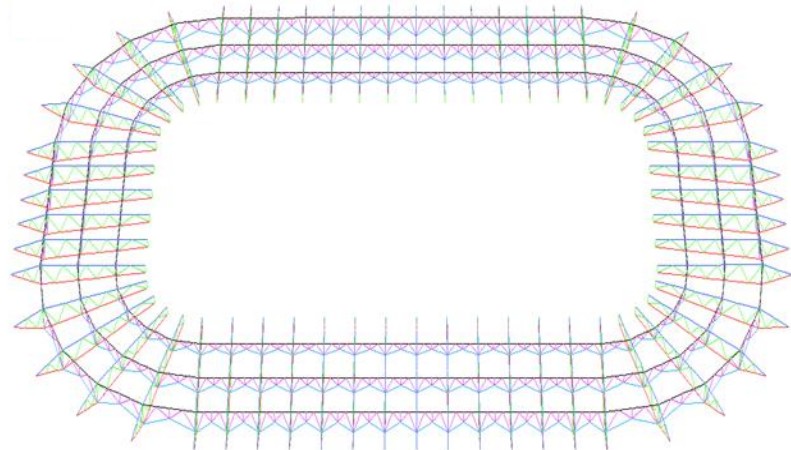

**Figure 5.** Three-Dimensional Isotropic View of CommBank Stadium.

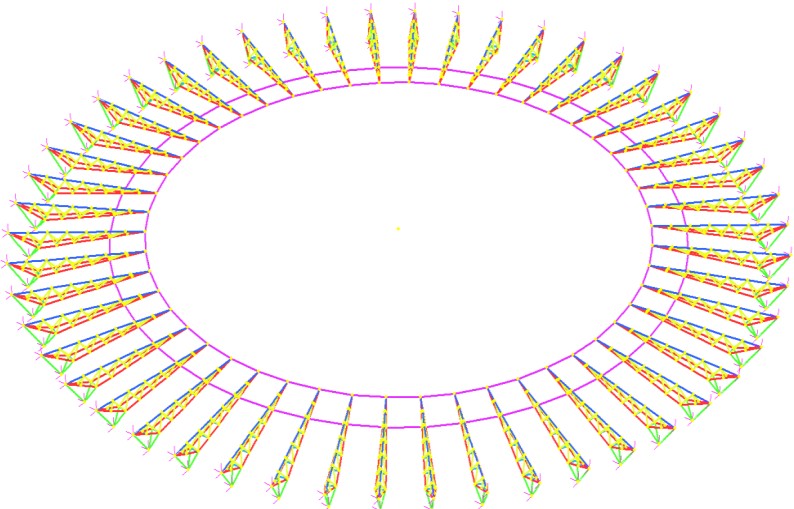

**Figure 6.** Three-Dimensional Isotropic View of Optus Stadium.

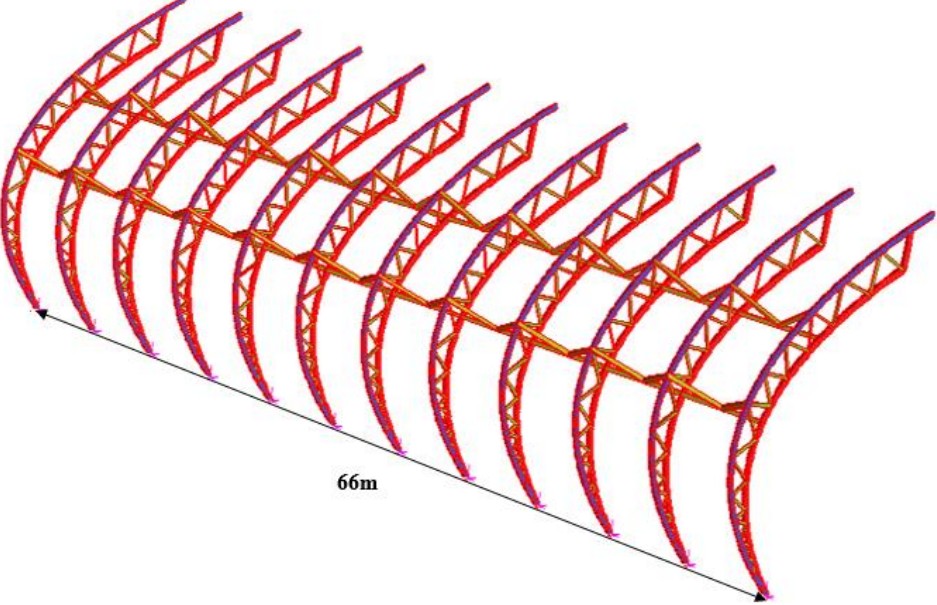

**Figure 7.** Three-Dimensional Isotropic View of Lakhwiya Stadium.

For Optus Stadium, the truss has a total length of 35 m, a maximum depth of 3 m, and a maximum width between bottom chords of 2.4 m [39–41]. The height from the top chord to the bottom of the truss is 6 m. The shape of the roof structure is a circle. The cylindrical coordinate is set up at the centre of the circular, which is 100 m away from the far end of the top chord. This bay of truss is then rotated by 7.2 degrees with a radius of 100 m fifty times to form the stadium's roof structure. Two truss bays are connected by two straight structural elements (i.e., purple members) at the bottom chords. All nodal connections can transfer axial force, shear and bending moment.

The methodology of modelling the Lakhwiya stadium will now be detailed. As the roof structure consisted of a series of arched cantilever trusses in combination with a polyester fabric roof [42–44], an individual member was first modelled. Since the members were parabolic-like and not linear, it was not feasible to model the structure in Strand7 using traditional methods of creating individual nodes and subsequently connecting them with beam elements [45]. Instead, the built-in tool "Points and Lines" was used to model the individual members where curvature is simulated by creating a number of smaller individual "steps" where a straight beam connects each step. The top chord of the truss was firstly modelled using the "points and lines" tool by determining and inputting three critical points, namely the bottom point, point of inflection or vertex, and the top point. In addition, 50 "steps" were created in order to increase the accuracy of the simulated curved member. It should be noted that an increase in steps will correspond to increased precision of modelled curved members. This was then repeated for the bottom chord and the diagonal members were drawn by connecting nodes from the top and bottom chord using "Beam2" elements. The restraint of the bottom nodes of all truss members was assumed to be fixed and therefore was restrained in the x, y, and z direction within the model.

Below, Figure 4 illustrates a 2D display of an individual truss member and its dimensions of being 23 m in height and a width of 16 m. As there were 7 bays, the singular truss member was then copied 8 times horizontally at 5.5 m intervals using the "copy by increment" function in Strand7 to create the final model. Figure 7 below displays the 3D model in isometric view. Finally, the triangular cross bracing was then created by connecting individual nodes with "Beam2" elements between each truss. It should also be noted that in some instances, the position of nodes was not centered on the top and bottom chord when creating the cross-bracing. As such, the corresponding member was subdivided to ensure the correct positioning of the cross-bracing.

### 3. Results

#### 3.1. Linear Static Analysis

The results of axial forces (Tension/Compression) and the bending moments of critical members under different load combinations in linear static analysis are imported and recorded in Excel. This was followed by calculating the tension/compression capacity and bending moment capacity to determine if the selected member size could withstand the critical loads/bending moments selected from the maximum value in different load combinations.

Tables 3–16 record the results of the linear static analyses of different load combinations for further investigation on the section/member capacity for all three stadiums. Note that the G + 0.7Q combination of the serviceability limit state is not shown in the linear static analysis results due to its combination load factors being smaller than G + 0.7Q + W which means if the structure behavior under G + 0.7Q + W combinations is satisfying the criteria and requirements, it will satisfy in G + 0.7Q as well.

The calculation of the tension/compression capacity and the bending moment capacity of different member sizes is conducted based on the geometries and material properties of OneSteel. As shown in Tables 17–25, the maximum result values will be compared with the theoretical capacity of member sizes selected at the beginning of the initial guess. Effective lengths of critical members should be calculated under AS4100:2020 [8] for compression

and bending moment capacity, while tension capacity could be directly calculated. For the mainframe group, the effective length of the top chord and bottom chord is the distance from the far end of the member to the nearest connecting frame section, while for the diagonal chord, the effective length is the length of the member itself. For the connection frame-group, the reference substance for the effective length is the main frame section for the top and bottom chords, which is adverse to the previous method above.

The procedure of comparison was conducted in Excel as well. The maximum result values were entered for comparison to check what types of member sizes are suitable for the current critical load combination. This procedure tries to push the member selection to the critical member size, which means selecting the most appropriate member size that just meets and satisfies all the capacity criteria within the linear static analysis. In this part, only the capacities of predetermined member sizes are considered to check if the guesses are established.

Note that the units of capacity are in kilonewton (kN) for force and kilonewton meter (kNm) for the moment.

**Table 3.** Main Frame of CommBank Stadium Roof (Axial force in kN).

| Load Combination | Top Chord | | Bot Chord | | Diagonal Chord | |
|---|---|---|---|---|---|---|
| | Tension | Comp. | Tension | Comp. | Tension | Comp. |
| 1.2G + 1.5Q | 523.9 | −14.6 | 154.7 | −629 | 490.8 | −461 |
| 1.35G | 148 | −1.25 | 44.61 | −181 | 139.6 | −131 |
| 1.2G + W | 16.6 | −420 | 470.4 | −118 | 313.3 | −386 |
| G + 0.7Q + W | 11.85 | −261 | 281.5 | −72 | 187.2 | −238 |
| **Maximum** | 523.9 | −420 | 470.4 | −629 | 490.8 | −461 |

For CommBank model, the members are subjected to experience tension and compression at the same time.

**Table 4.** Connection Frame of CommBank Stadium Roof (Axial force in kN).

| Load Combination | Top Chord | | Bot Chord | | Diagonal Chord | |
|---|---|---|---|---|---|---|
| | Tension | Comp. | Tension | Comp. | Tension | Comp. |
| 1.2G + 1.5Q | 0.62 | −194 | 33.5 | −67.4 | 27.2 | −36.2 |
| 1.35G | 0.9 | −54.7 | 8.05 | −20.7 | 8.09 | −8.92 |
| 1.2G + W | 123.3 | −1.16 | 35.3 | −35.7 | 22.8 | −18.3 |
| G + 0.7Q + W | 82.1 | −9.84 | 30.3 | −37.2 | 19.8 | −19.2 |
| **Maximum** | 123.3 | −194 | 35.3 | −37.2 | 27.2 | −36.2 |

For CommBank model, the members are subjected to experience tension and compression at the same time.

**Table 5.** Main Frame of CommBank Stadium Roof (Bending in kNm).

| Load Combination | Top Chord | | Bot Chord | | Diagonal Chord | |
|---|---|---|---|---|---|---|
| | End1 | End2 | End1 | End2 | End1 | End2 |
| 1.2G + 1.5Q | 0.341 | 3.84 | 0.376 | 19.7 | 1.24 | 9.6 |
| 1.35G | 0.045 | 0.8 | 0.03 | 2.07 | 0.33 | 1.51 |
| 1.2G + W | 0.336 | 4.31 | 0.28 | 48.72 | 0.47 | 28.92 |
| G + 0.7Q + W | 0.2 | 3.05 | 0.15 | 34.06 | 0.22 | 20.66 |
| **Maximum** | 0.341 | 4.31 | 0.376 | 48.72 | 1.24 | 28.92 |

**Table 6.** Connection Frame of CommBank Stadium Roof (Bending in kNm).

| Load Combination | Top Chord | | Bot Chord | | Diagonal Chord | |
|---|---|---|---|---|---|---|
| | End1 | End2 | End1 | End2 | End1 | End2 |
| 1.2G + 1.5Q | 0.73 | 8.7 | 1.22 | 1.67 | 0.55 | 2.18 |
| 1.35G | 0.13 | 2.31 | 1.35 | 1.65 | 0.57 | 0.78 |
| 1.2G + W | 0.98 | 5.09 | 1.21 | 1.47 | 0.53 | 3.27 |
| G + 0.7Q + W | 0.72 | 2.33 | 1 | 1.16 | 0.43 | 2.41 |
| **Maximum** | 0.98 | 8.7 | 1.35 | 1.67 | 0.57 | 3.27 |

**Table 7.** Central Frame of Lakhwiya Stadium (Tensile Force in kN).

| Load Combination | Top Chord | Bot Chord | Diagonal Chord | Cross-Connections |
|---|---|---|---|---|
| 1.2G + 1.5Q | 585 | n/a | 103 | 76 |
| 1.35G | 156 | n/a | 23.6 | 0.56 |
| 1.2G + W | n/a | 2244 | 448 | 21 |
| G + 0.7Q + W | n/a | 2017.19 | 401 | 16 |
| **Maximum** | 156 | 2265 | 456 | 76 |

**Table 8.** Central Frame of Lakhwiya Stadium (Compressive Force kN).

| Load Combination | Top Chord | Bot Chord | Diagonal Chord | Cross-Connections |
|---|---|---|---|---|
| 1.2G + 1.5Q | 0.36 | 611 | 128 | 68 |
| 1.35G | 0.31 | 179 | 45 | 5 |
| 1.2G + W | 2169 | n/a | 404 | 22 |
| G + 0.7Q + W | 1951 | n/a | 363 | 17 |
| **Maximum** | 2193 | 611 | 415 | 68 |

**Table 9.** Central Frame of Lakhwiya Stadium (Bending Force kN).

| Load Combination | Top Chord | Bot Chord | Diagonal Chord | Cross-Connections |
|---|---|---|---|---|
| 1.2G + 1.5Q | 50 | 57 | 0.31 | 0.30 |
| 1.35G | 0.42 | 0.40 | n/a | 0.085 |
| 1.2G + W | 12 | 14 | 0.74 | 0.65 |
| G + 0.7Q + W | 10.00 | 11 | 0.59 | 0.52 |
| **Maximum** | 50 | 57 | 2.4 | 0.65 |

**Table 10.** End Frame of Lakhwiya Stadium (Tensile Force in kN).

| Load Combination | Top Chord | Bot Chord | Diagonal Chord | Cross-Connections |
|---|---|---|---|---|
| 1.2G + 1.5Q | 446.36 | n/a | 78.24 | 76 |
| 1.35G | 143.63 | n/a | 21.97 | 0.56 |
| 1.2G + W | n/a | 1886.90 | 381.77 | 21 |
| G + 0.7Q + W | n/a | 1734.83 | 349.74 | 16.36 |
| **Maximum** | 446.36 | 1906 | 389 | 76 |

**Table 11.** End Frame of Lakhwiya Stadium (Compressive Force in kN).

| Load Combination | Top Chord | Bot Chord | Diagonal Chord | Cross-Connections |
|---|---|---|---|---|
| 1.2G + 1.5Q | 0.36 | 465.35 | 98.53 | 68 |
| 1.35G | 0.31 | 164.4 | 41.07 | 5.05 |
| 1.2G + W | 1819.5 | n/a | 341.93 | 22 |
| G + 0.7Q + W | 1673.13 | n/a | 313.79 | 17.38 |
| **Maximum** | 1842 | 465.35 | 341.93 | 68 |

**Table 12.** End Frame of Lakhwiya Stadium (Bending Force kN).

| Load Combination | Top Chord | Bot Chord | Diagonal Chord | Cross-Connections |
|---|---|---|---|---|
| 1.2G + 1.5Q | 38.82 | 52.47 | 0.42 | 0.3 |
| 1.35G | −0.5 | 0.78 | 0.0052 | 0.085 |
| 1.2G + W | 10.16 | 13.31 | 1.08 | 0.65 |
| G + 0.7Q + W | 8.04 | 10.46 | 0.80 | 0.52 |
| **Maximum** | 38.82 | 52.47 | 3.03 | 0.65 |

**Table 13.** Main Frame of Optus Stadium (Tensile Force in kN).

| Load Combination | Top Chord | Bot Chord | Web Diagonal | Bot Diagonal | Bot Structure |
|---|---|---|---|---|---|
| 1.2G + 1.5Q | 208.2 | n/a | 85.8 | 19.5 | 3.4 |
| 1.35G | 432.6 | n/a | 27.2 | 17.5 | 3.9 |
| 1.2G + W | 453.5 | 51.5 | 39.4 | 16.7 | 10.7 |
| G + 0.7Q + W | 834.5 | n/a | 64.9 | 34 | 2.9 |
| **Maximum** | 834.5 | 51.5 | 85.8 | 19.5 | 10.7 |

**Table 14.** Main Frame of Optus Stadium (Compressive Force in kN).

| Load Combination | Top Chord | Bot Chord | Web Diagonal | Bot Diagonal | Bot Structure |
|---|---|---|---|---|---|
| 1.2G + 1.5Q | n/a | −615.4 | −97.3 | −21.8 | −458 |
| 1.35G | n/a | −202.5 | −33 | −6.4 | −154 |
| 1.2G + W | −102.8 | −205.6 | −41.5 | −6 | −173 |
| G + 0.7Q + W | n/a | −380 | −66.9 | −10.8 | −301 |
| **Maximum** | −102.8 | −615.4 | −97.3 | −21.8 | −301 |

**Table 15.** Main Frame of Optus Stadium (Bending Moment in kNm).

| Load Combination | Top Chord | Bot Chord | Web Diagonal | Bot Diagonal | Bot Structure |
|---|---|---|---|---|---|
| 1.2G + 1.5Q | 17.8 | 40.3 | 2.8 | 0.17 | 35.2 |
| 1.35G | 7.7 | 11.6 | 1.2 | 0.17 | 11.7 |
| 1.2G + W | 7.5 | 14.6 | 0.9 | 0.16 | 12.1 |
| G + 0.7Q + W | 11.9 | 25.6 | 1.9 | 0.14 | 21.8 |
| **Maximum** | 17.8 | 40.3 | 2.8 | 0.17 | 35.2 |

**Table 16.** Connections of Optus Stadium.

| Load Combination | Tensile Force kN | Compressive Force kN | Bending Moment kNm |
|---|---|---|---|
| 1.2G + 1.5 | n/a | −30.5 | 0.38 |
| 1.35G | n/a | −9.8 | 0.44 |
| 1.2G + W | n/a | −5 | 0.88 |
| G + 0.7Q + W | n/a | −13.8 | 0.8 |
| **Maximum** | n/a | −30.5 | 0.88 |

**Table 17.** Tensile Capacity vs. Critical Loading (CommBank Stadium).

| Member | Member Size | Tensile Capacity | Critical Loading |
|---|---|---|---|
| Main Top Chord | C450L0 150 × 150 × 6 | 1031.71 | 523.9 |
| Main Bot Chord | C450L0 150 × 150 × 8 | 1338.44 | 470.4 |
| Main Diag. Chord | C450L0 150 × 150 × 5 | 870.61 | 490.8 |
| Conn. Top Chord | C450L0 125 × 125 × 5 | 715.7 | 123.3 |
| Conn. Bot Chord | C450L0 125 × 125 × 6 | 845.8 | 35.3 |
| Conn. Diag. Chord | C450L0 125 × 125 × 4 | 582.47 | 27.2 |

**Table 18.** Compressive Capacity vs. Critical Loading (CommBank Stadium).

| Member | Member Size | Compressive Capacity | Critical Loading |
|---|---|---|---|
| Main Top Chord | C450L0 150 × 150 × 6 | 1266.62 | 420 |
| Main Bot Chord | C450L0 150 × 150 × 8 | 1373.34 | 628.9 |
| Main Diag. Chord | C450L0 150 × 150 × 5 | 849.07 | 460.6 |
| Conn. Top Chord | C450L0 125 × 125 × 5 | 863.2 | 193.5 |
| Conn. Bot Chord | C450L0 125 × 125 × 6 | 865.8 | 37.2 |
| Conn. Diag. Chord | C450L0 125 × 125 × 4 | 673.06 | 36.2 |

**Table 19.** Bending Moment Capacity vs. Critical Loading (CommBank Stadium).

| Member | Member Size | Bending Moment Capacity | Critical Moment |
|---|---|---|---|
| Main Top Chord | C450L0 150 × 150 × 6 | 60.75 | 4.31 |
| Main Bot Chord | C450L0 150 × 150 × 8 | 76.14 | 48.72 |
| Main Diag. Chord | C450L0 150 × 150 × 5 | 42.86 | 28.92 |
| Conn. Top Chord | C450L0 125 × 125 × 5 | 29.66 | 8.7 |
| Conn. Bot Chord | C450L0 125 × 125 × 6 | 40.91 | 1.67 |
| Conn. Diag. Chord | C450L0 125 × 125 × 4 | 29.28 | 3.27 |

**Table 20.** Tensile Capacity vs. Critical Loading (Lakhwiya Stadium).

| Member | Member Size | Tensile Capacity | Critical Loading |
|---|---|---|---|
| Central Top Chord | C450L0 200 × 200 × 8 | 1834 | 585 |
| Central Bot Chord | C450L0 200 × 200 × 9 | 2265 | 2045 |
| Central Diag Chord | C450L0 100 × 100 × 4 | 459 | 456 |
| Central Cross-bracing | C450L0 65 × 65 × 3 | 223.4 | 21 |
| End Top Chord | C450L0 200 × 200 × 8 | 1834 | 446 |
| End Bot Chord | C450L0 200 × 200 × 9 | 2045 | 1906 |
| End Diag Chord | C450L0 65 × 65 × 6 | 400 | 389 |
| End Cross-bracing | C450L0 65 × 65 × 3 | 223.4 | 21 |

**Table 21.** Compressive Capacity vs. Critical Loading (Lakhwiya Stadium).

| Member | Member Size | Compressive Capacity | Critical Loading |
|---|---|---|---|
| Central Top Chord | C450L0 200 × 200 × 8 | 2285 | 2169 |
| Central Bot Chord | C450L0 200 × 200 × 9 | 1810 | 611 |
| Central Diag Chord | C450L0 100 × 100 × 4 | 534 | 404 |
| Central Cross-bracing | C450L0 65 × 65 × 3 | 26 | 22 |
| End Top Chord | C450L0 200 × 200 × 8 | 2285 | 1842 |
| End Bot Chord | C450L0 200 × 200 × 9 | 1810 | 465 |
| End Diag Chord | C450L0 65 × 65 × 6 | 356 | 351 |
| End Cross-bracing | C450L0 65 × 65 × 3 | 26 | 22 |

**Table 22.** Bending Moment Capacity vs. Critical Loading (Lakhwiya Stadium).

| Member | Member Size | Bending Capacity | Critical Loading |
|---|---|---|---|
| Central Top Chord | C450L0 200 × 200 × 8 | 112 | 50 |
| Central Bot Chord | C450L0 200 × 200 × 9 | 123 | 57 |
| Central Diag Chord | C450L0 100 × 100 × 4 | 14 | 2.4 |
| Central Cross-bracing | C450L0 65 × 65 × 3 | 5.2 | 0.65 |
| End Top Chord | C450L0 200 × 200 × 8 | 112 | 38.82 |
| End Bot Chord | C450L0 200 × 200 × 9 | 123 | 52.47 |
| End Diag Chord | C450L0 65 × 65 × 6 | 7 | 3 |
| End Cross-bracing | C450L0 65 × 65 × 3 | 5.2 | 0.65 |

**Table 23.** Tensile Capacity vs. Critical Loading (Optus Stadium).

| Member | Member Size | Tensile Capacity | Critical Loading |
|---|---|---|---|
| Top Chord | C350L0 273.1 × 12.7 | 3143 | 834.5 |
| Bot Chord | C350L0 273.1 × 12.7 | 3143 | 51.5 |
| Web Diagonal | C350L0 101.6 × 3.2 | 311 | 85.8 |
| Bot Diagonal | C350L0 101.6 × 3.2 | 311 | 34 |
| Bot Structure | C350L0 168.3 × 7.1 | 1134 | 10.7 |
| Connections | C350L0 88.9 × 2.6 | 222 | n/a |

**Table 24.** Compressive Capacity vs. Critical Loading (Optus Stadium).

| Member | Member Size | Compressive Capacity | Critical Loading |
|---|---|---|---|
| Top Chord | C350L0 273.1 × 12.7 | −3143 | −102.8 |
| Bot Chord | C350L0 273.1 × 12.7 | −3143 | −615.4 |
| Web Diagonal | C350L0 101.6 × 3.2 | −161 | −97.3 |
| Bot Diagonal | C350L0 101.6 × 3.2 | −161 | −21.8 |
| Bot Structure | C350L0 168.3 × 7.1 | 961 | −458 |
| Connections | C350L0 88.9 × 2.6 | −91.3 | −30.5 |

**Table 25.** Bending Moment Capacity vs. Critical Loading (Optus Stadium).

| Member | Member Size | Bending Moment Capacity | Critical moment |
|---|---|---|---|
| Top Chord | C350L0 273.1 × 12.7 | 263 | 17.8 |
| Bot Chord | C350L0 273.1 × 12.7 | 263 | 40.3 |
| Web Diagonal | C350L0 101.6 × 3.2 | 6.54 | 2.8 |
| Bot Diagonal | C350L0 101.6 × 3.2 | 6.54 | 0.17 |
| Bot Structure | C350L0 168.3 × 7.1 | 53 | 35.2 |
| Connections | C350L0 88.9 × 2.6 | 3.28 | 0.88 |

As shown in Tables 17–25 above, the initial guess of member selections in different parts of the structure all satisfy the design capacity and standard requirements in critically applied load combinations. However, the capacity is far over the critical applied loading. Hence, as mentioned above, the comparison between the critical load cases and different member sizes could push the selection to the most critical member size. Figure 8 below shows and summarises the deformed shape under 1.2G + W combination in Linear static analysis with a 5% displacement scale of three stadium roof models.

| Load Comb. | CommBank Stadium | Optus Stadium | Lakhwiya Stadium |
|---|---|---|---|
| 1.2G + W | | | |

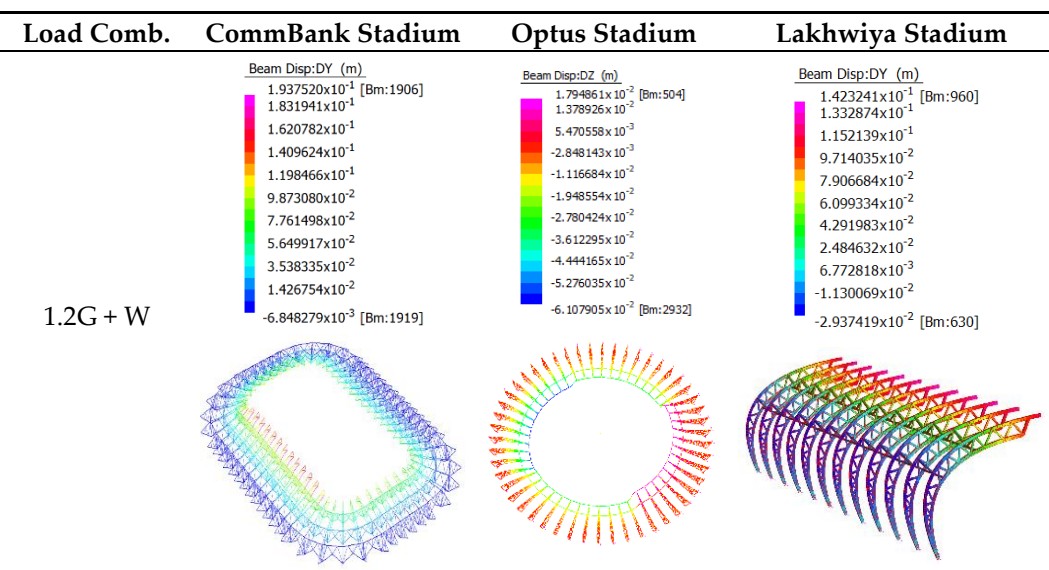

**Figure 8.** Deformed shape of CommBank, Optus, and Lakhwiya Stadium under 1.2G + W with 5% displacement scale.

### *3.2. Linear Buckling Analysis Result*

When conducting the linear buckling, the initial condition is the load combination that is selected. Numbers of mode shape could be selected as well. In this model, only four modes are selected to conduct the linear buckling solver. Below, Figures 9–11 show the results of the First Mode shapes of deformation under linear buckling analysis in the 1.2G + W combination of three stadium roof models and their correspondent eigenvalues.

Table 26 concludes the eigenvalues of the First Mode shape of deformations in the three stadium roof models.

**Table 26.** Eigenvalue of First Mode shape of deformation.

| Load Combination | CommBank Stadium Eigenvalue 1 | Optus Stadium Eigenvalue 1 | Lakhwiya Stadium Eigenvalue 1 |
|---|---|---|---|
| 1.2G + 1.5Q | 3.67 | 5.76 | 22.34 |
| 1.35G | 13.91 | 18.5 | 46.62 |
| 1.2G + W | 6.17 | 21.4 | 8.72 |
| G + 0.7Q + W | 7.89 | 11.2 | 9.6 |

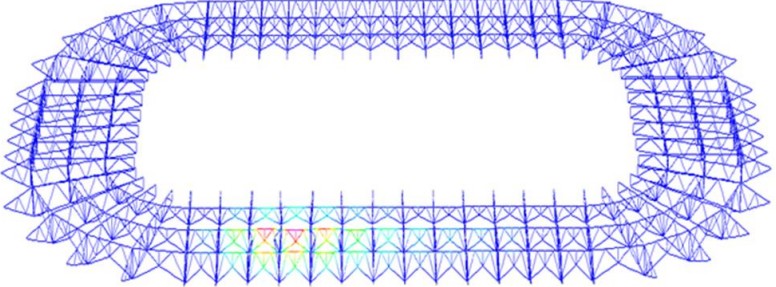

**Figure 9.** CommBank stadium First Mode deformation shape under linear buckling analysis in 1.2G + W.

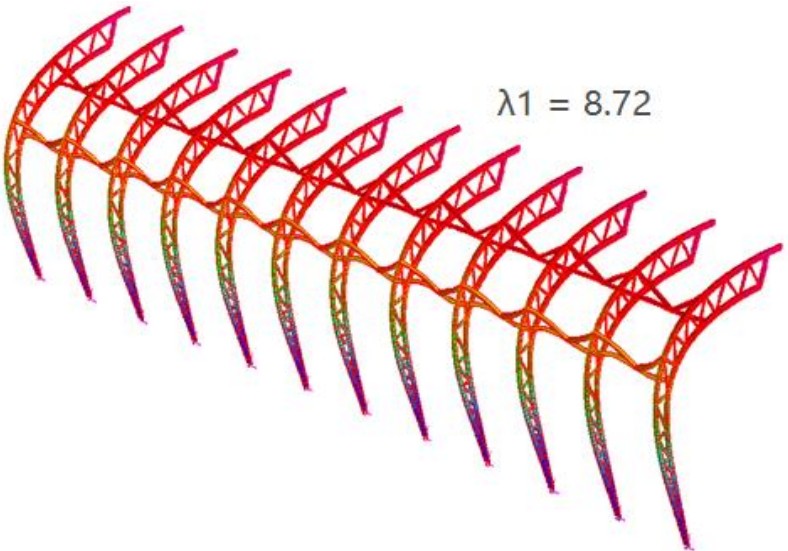

**Figure 10.** Lakhwiya Stadium First Mode deformation shape under linear buckling analysis in 1.2G + W.

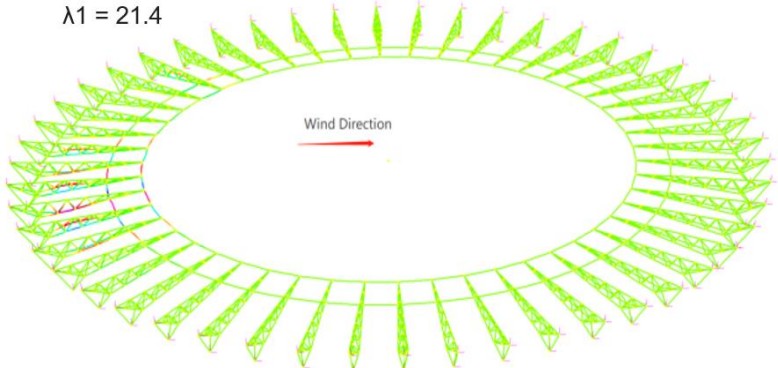

**Figure 11.** Optus Stadium First Mode deformation shape under linear buckling analysis in 1.2G + W.

### 3.3. Nonlinear Buckling Analysis Result

In the nonlinear buckling analysis dialog, both Geometry nonlinearity (GNL) and Material nonlinearity (MNL) are selected to conduct the analysis. Sub-increments are set with arc length controls as well. To clarify, the sub-increments could be set based on the eigenvalues from the linear buckling analysis as linear buckling analysis overestimates the buckling capacity of the member (Please see Figures 12–14). After conducting the nonlinear research, the load factor vs. displacement data points were plotted to illustrate the buckling behavior under different load factors of different load combinations. This is shown below in Figures 15–20. The eigenvalues from the linear buckling analysis were compared with the load factor from nonlinear buckling analysis to investigate and discuss the difference between the two analysis methods [45]. Note that only the buckling deformations of the stadium roof models in 1.2G + W are shown below in the figures as illustration examples.

Buckling Load factor = 4.29

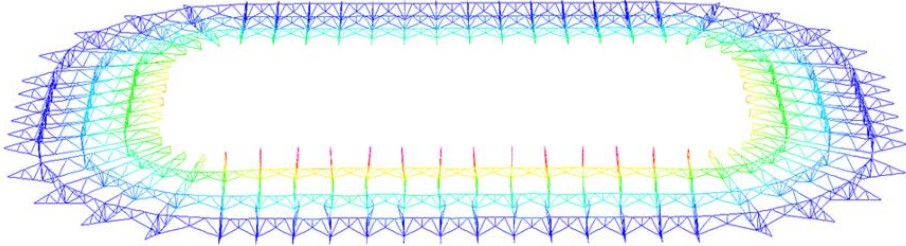

**Figure 12.** CommBank Stadium deformation shape under non-linear buckling analysis in 1.2G + W.

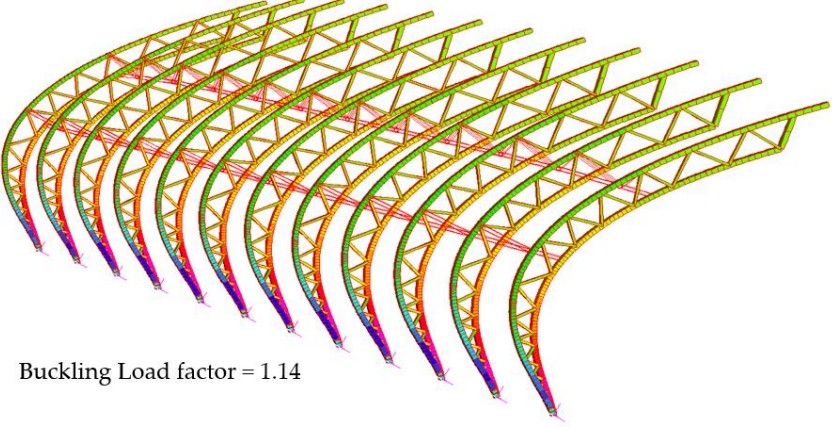

Buckling Load factor = 1.14

**Figure 13.** Lakhwiya Stadium deformation shape under non-linear buckling analysis in 1.2G + W.

Buckling Load factor = 9.03

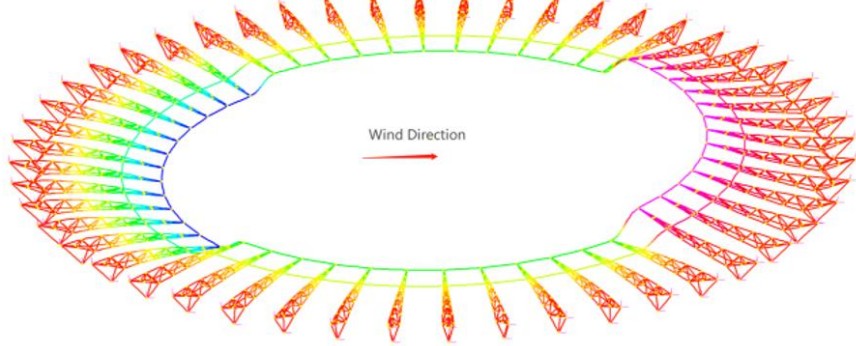

**Figure 14.** Optus Stadium deformation shape under non-linear buckling analysis in 1.2G + W.

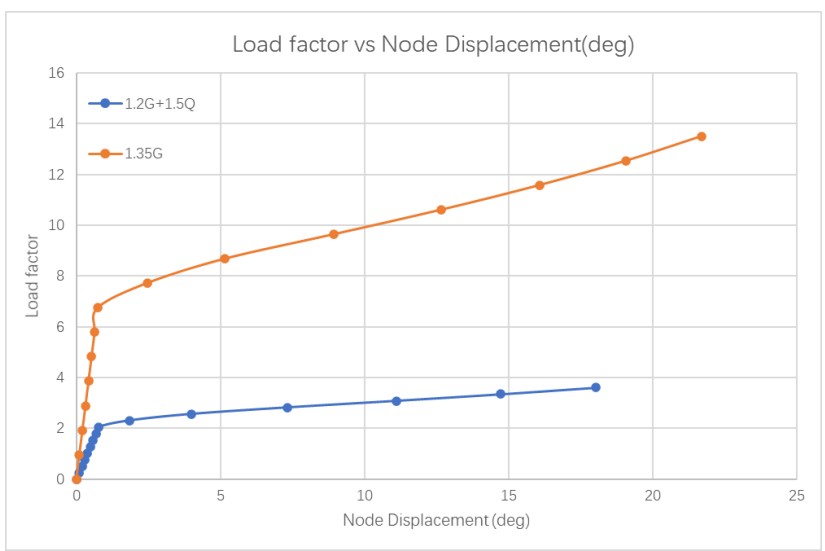

**Figure 15.** CommBank Stadium Load Factor vs. Nodal Rotation (dead and live load cases).

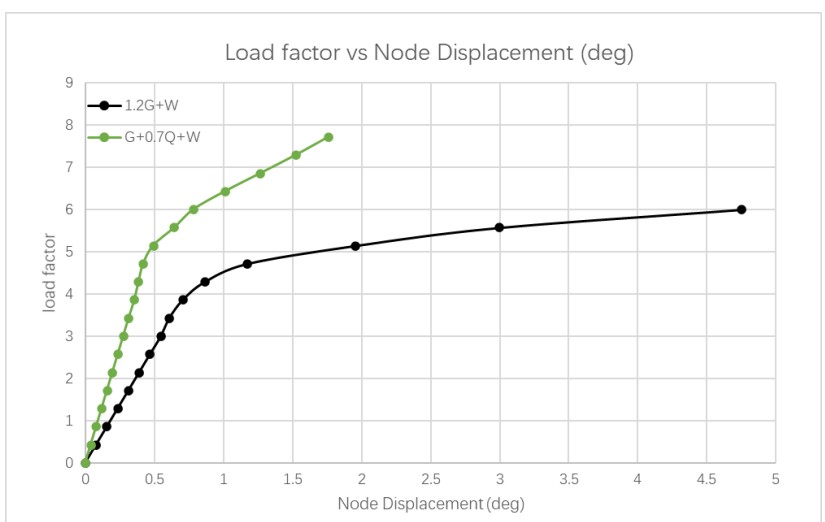

**Figure 16.** CommBank Stadium Load Factor vs. Nodal Rotation (wind load cases).

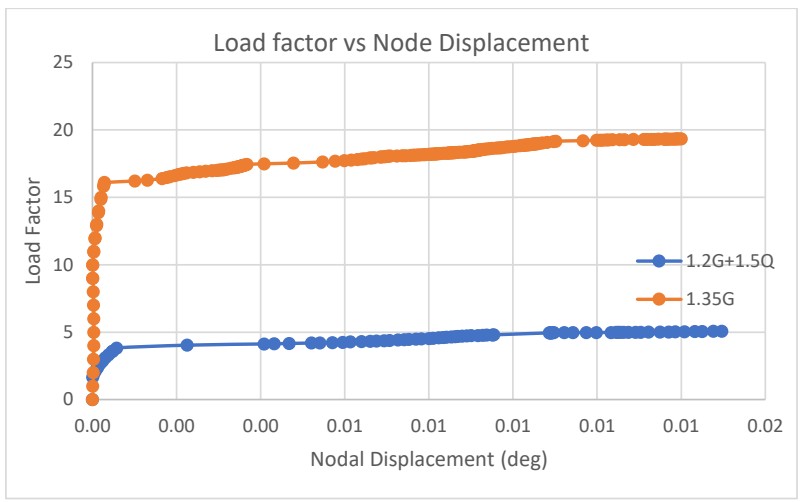

**Figure 17.** Lakhwiya Stadium Load Factor vs. Nodal Displacement (dead and live load cases).

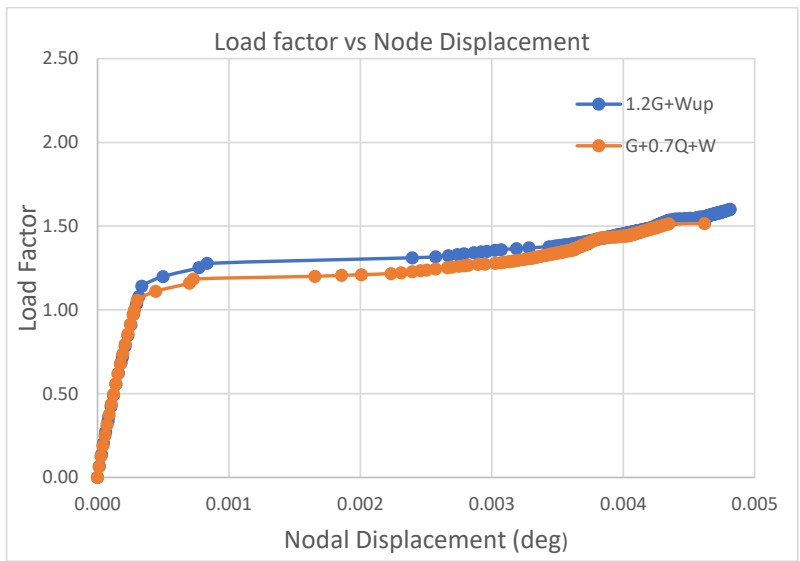

**Figure 18.** Lakhwiya Stadium Load Factor vs. Nodal Displacement (wind load cases).

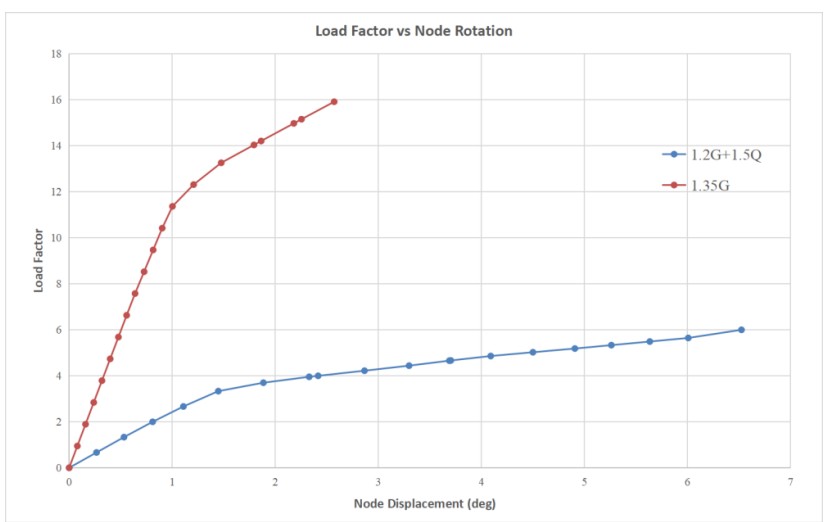

**Figure 19.** Optus Stadium Load Factor vs. Nodal Rotation (dead and live load cases).

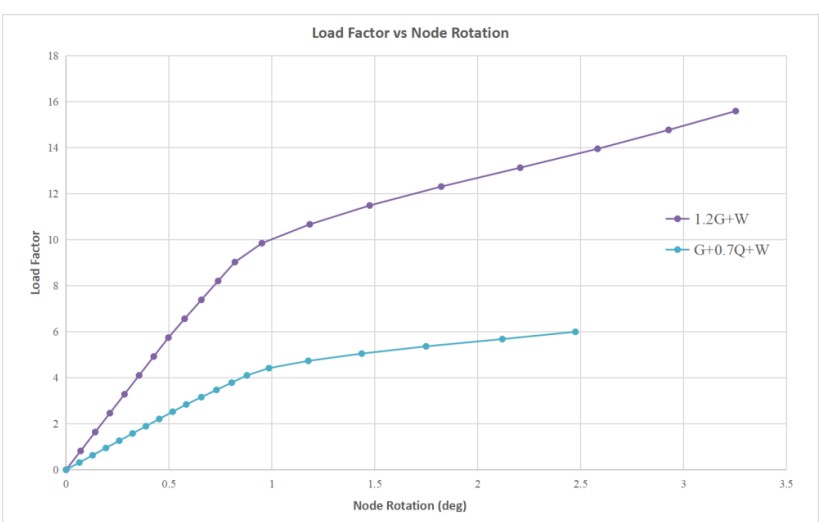

**Figure 20.** Optus Stadium Load Factor vs. Nodal Rotation (wind load cases).



In order to gain a deeper insight into the nonlinear analysis result, the displacement and rotation of a particular node in the nonlinear analysis model for all load combinations are demonstrated in Figures 15–20. The displacement and rotation are plotted against the load factor in scatter plots. The corresponding data are exported from Strand7 to Excel to generate the plot. The load cases are separated into two categories: wind involved and wind free. For load increments that cannot converge under the iteration limit directly, sub-increments are automatically added to try to reach the convergence in Strand7.

By observing the graphs presented in Figures 15–20, the critical buckling load factor of the nonlinear analysis for each case can be obtained at the position where the gradient changes. The graphs exhibit a linear relationship between the load factor and the node displacement or node rotation in the beginning. The gradient of the line for each load case becomes gradual after reaching the critical buckling load. Either post-buckling collapse (i.e., the downward curve after the bifurcation point) or post-buckling (i.e., the secondary path) will occur when the load reaches the critical buckling load. If a post-buckling collapse happens, a snapback will occur in the load factor vs. the node displacement graph. The member's strength will decrease quickly, and the displacement will increase significantly. In this model, the snapback is not observed in the above graphs, implying that the post-buckling collapse has not occurred. Alternatively, the graphs follow the secondary path after the bifurcation point. In this case, the member can still withstand the increased load, but the structure displacement will increase. When the structure starts to buckle, it will not immediately collapse. The significant deflection of the structural members can detect damage. Therefore, the design is considered ductile, and the structural failure will not cause a disaster in a short period.

The Tables 27–29 summarise the load factors and eigenvalues of different models and their respective critical loads under different load combinations for further comparison.

**Table 27.** CommBank Stadium Linear vs. Non-linear Buckling.

| Load Combination | Eigenvalue (LB) | Critical Load (LB) | Load Factor (NLB) | Critical Load (NLB) |
|---|---|---|---|---|
| 1.2G + 1.5Q | 3.67 | 363.84 | 2.06 | 204.23 |
| 1.35G | 13.91 | 161.91 | 6.75 | 78.86 |
| 1.2G + W | 6.17 | 942.09 | 4.29 | 655.04 |
| G + 0.7Q + W | 7.89 | 1895.10 | 5.14 | 1234.57 |

**Table 28.** Lakhwiya Stadium Linear vs. Non-linear Buckling.

| Load Combination | Eigenvalue (LB) | Critical Load (LB) | Load Factor (NLB) | Critical Load (NLB) |
|---|---|---|---|---|
| 1.2G + 1.5Q | 22.34 | 817 | 3.83 | 165.84 |
| 1.35G | 46.62 | 445 | 15.83 | 151.10 |
| 1.2G + W | 8.72 | 1219 | 1.14 | 155 |
| G + 0.7Q + W | 9.6 | 1179 | 1.06 | 188 |

**Table 29.** Optus Stadium Linear vs. Non-linear Buckling.

| Load Combination | Eigenvalue (LB) | Critical Load (LB) | Load Factor (NLB) | Critical Load (NLB) |
|---|---|---|---|---|
| 1.2G + 1.5Q | 5.76 | 980 | 3.33 | 566 |
| 1.35G | 18.5 | 669 | 11.4 | 431 |
| 1.2G + W | 21.4 | 1532 | 9.03 | 647 |
| G + 0.7Q + W | 11.2 | 1453 | 4.1 | 532 |

Note that the unit of critical loading is in kilonewton (kN).

## 4. Discussion

In terms of the linear static analysis, the self-weight of the structural member, live load, and wind load are calculated and input to Strand7. The five load combinations that are required to be checked as outlined in AS1170 [16] are analysed. Calculations of the

critical axial tension, axial compression, bending moment, and combined actions are made; the comparison between the analysis result and the calculation shows that the design has sufficient strength. It was concluded that axial forces govern the design for the truss members which was within expectations as, theoretically, members would exclusively experience axial force [46]. In addition, the critical load case under linear static analysis involved wind loads. This is because stadiums often have long spans and increased heights, which will lead to increased wind velocities, which correlates to exponentially increasing wind loads [47,48]. Linear buckling analysis indicated that the critical loading factors were significantly lower for wind load cases. However, as the applied wind forces were higher than structural self-weight, dead, and live loads, the critical buckling load was also higher for the wind load. However, by examining the obtained nominal displacement of the structure, it was found that under wind loading, the structure may run into a minor deformation compared with other cases.

Nonlinear buckling analysis confirmed the theoretical framework of the overestimation of the critical buckling load factor by linear buckling analysis. Under non-linear buckling analysis, the critical buckling load was reduced by up to 90% for cases involving wind which indicated the presence of considerable geometric and material non-linearity of the structure [49]. In addition, the critical buckling load for the wind load cases approached closer to one, at which point the existing load combination applied on the structure would cause buckling to occur. However, after the bifurcation point is reached, the structure will not experience post-buckling collapse [50,51]. Instead, the structure will follow the secondary post-buckling path and fail under serviceability conditions which is desirable as opposed to sudden collapse. Moreover, examining the vertical displacement of the roofing structure, linear analysis also underestimated the maximum displacements in comparison with the non-linear analysis by up to 20 times.

Furthermore, during the initial phase of defining loading cases, difficulties arose involving the determination of the aerodynamic factor of the structural shape due to the high variations and uncertainties in aerodynamic behaviour [39] If resources permit, a wind tunnel simulation or simulation via CFD programs (computational fluid dynamics) software should be conducted [39]. In light structures, the wind load often governs the design. Therefore, simulating the effect of the wind loads on the critical sections may result in a cost-effective outcome.

Below, Figure 21 displays a design flow that aims to provide a universal design procedure for stadium roofing structures using finite element analysis. The flowchart exhibits a universal design process whereby civil engineers can apply regardless of architectural design, material, and structure choice (truss or beam). The design process considers failure modes under elastic material properties through linear analysis and nonlinear properties such as geometric and material non-linearity through advanced analysis.

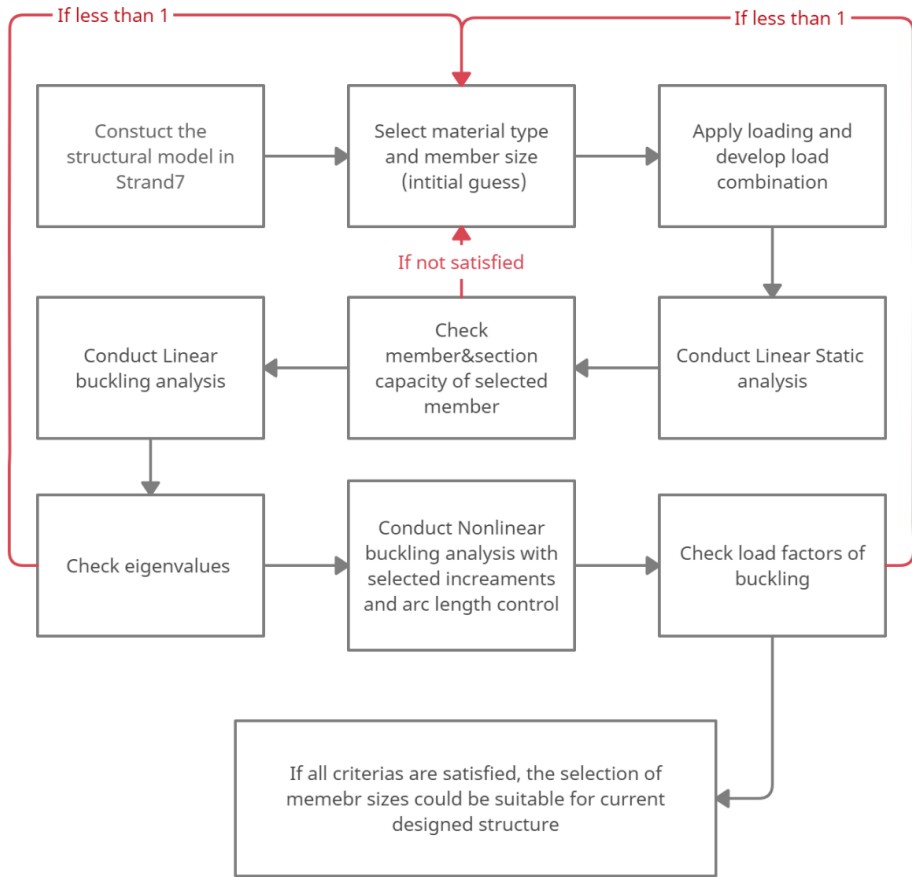

**Figure 21.** Universal design flowchart for stadium roofing structure.

## 5. Conclusions

This paper aims to construct a design procedure that will allow for the universal application by civil engineers for designing stadium roofing structures irrespective of critical factors. This was achieved by using the CommBank, Optus, and Lakhwiya stadiums as experimental cases, where basic preliminary member sizing and design feasibility checks were conducted using the linear static solver of the finite element analysis software Strand7. Under this section, tensile force, compressive force, flexural force, and deflection checks were conducted under the Australian steel structures code AS4100:2020 [8]. Linear buckling analysis based on a predefined stress–strain curve was then conducted. The linear buckling load factor was determined by inspecting the "mode 1" eigenvalue, and the critical buckling load was calculated. Based on the "mode 1" shape of the linear buckling analysis, non-linear buckling analysis was then conducted and through the assessment of the load factor vs. nodal displacement graph, the critical buckling load factor was then determined. Comparisons were then made between the linear buckling results and non-linear buckling results.

Additionally, the detailed design of the connection (i.e., bolting or welding) between structural members is not included, and it is suggested to analyse the impact of different connections on the internal actions induced in the structural member under various load cases. In reality, devices such as lights, speakers, and the digital screen will be installed and attached to the roof structure; it is necessary to determine the influence of extra loads from the devices to the roof in structural analysis.

Finally, a flow chart that illustrates the design procedure was presented above. The significant findings of this paper are presented below:

1.  Asymmetrical and innovative architectural design of modern stadium roofing structures results in difficulty when determining wind load using the Australian design code for wind action (AS1170.2), particularly when calculating aerodynamic factors.

2. Wind-tunnel simulation or computational fluid dynamics modelling should be undertaken during the design process of the stadium roofing structure in order to obtain accurate aerodynamic factors and precise wind-load application on a finite element analysis model.

3. Load combinations involving wind loads result in the largest axial forces and bending in the critical members when conducting linear analysis and governing member design.

4. Load combinations involving wind load results in the most significant deflections in critical members when conducting linear static analysis.

5. Structural self-weight often acts as a counteracting force to the critical load case of wind, and therefore heavier structures may be less susceptible to deflection failure via wind load.

6. Linear buckling analysis indicated that, under wind load combinations, minimal structural deformation compared with load cases involving dead and live load will occur at the critical buckling point.

**Author Contributions:** Conceptualization, F.T., E.C., A.H. and J.L.; Methodology, F.T., E.C., A.H. and J.L.; Software, F.T., E.C., A.H. and J.L.; Validation, F.T., E.C., A.H. and J.L.; Formal analysis, F.T., E.C., A.H. and J.L.; Investigation, F.T., E.C., A.H. and J.L.; Resources, F.T., E.C., A.H. and J.L. Data curation, F.T., E.C., A.H. and J.L.; Writing—review & editing, F.T., E.C., A.H. and J.L.; Project administration, F.T., E.C., A.H. and J.L. All authors have read and agreed to the published version of the manuscript.

**Funding:** This research received no external funding.

**Institutional Review Board Statement:** Not applicable.

**Informed Consent Statement:** Not applicable.

**Data Availability Statement:** Not applicable.

**Conflicts of Interest:** The authors declare no conflict of interest.

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
