# Peer review of "Designing Lightweight Stadium Roofing Structures Based on Advanced Analysis Methods"

_sustainability, doi:10.3390/su15043612_

Round 1

Reviewer 1 Report

Strange is the form of the abstract – the use of the future tense. It is not known why the Authors wrote the abstract in this way, after all they have already done the activities described there...

In item 20 of the bibliography there is no name of the Author of the cited website – it should be: „20. Skotny Ł. (2021, November 10). Linear Buckling in plain language. Enterfea. Retrieved October 25, 2022, from https://enterfea.com/linear-buckling-explained/”, because the Author of the website is Dr. Łukasz Skotny. In addition, no 20 and no 51 are one publication (it's the same website!).

The whole list of literature should be redrafted, because right now there is chaos.

In my opinion, items 20 and 28 (websites) are not sources to which Authors of scientific papers should refer, the same as the websites of companies offering commercial software for structural calculations. I am not trying to question the credibility of the content, but it cannot be the basis of scientific studies.

In line 168 should be “[22]” instead of “[22”.

Figure 2.2.2.1. needs to be amended – parts (a) and (b) should be harmonised!

Figure 6.1 also contains errors – already in the first step of “design flowchart” is written “Constuct” and should be “Construct”.

The entire manuscript must undergo a thorough editorial revision.

Author Response

Reviewer’s comments

Authors’ reply

Strange is the form of the abstract – the use of the future tense. It is not known why the Authors wrote the abstract in this way, after all they have already done the activities described there..

The abstracted was completely revised.

In item 20 of the bibliography there is no name of the Author of the cited website – it should be: „20. Skotny Ł. (2021, November 10). Linear Buckling in plain language. Enterfea. Retrieved October 25, 2022, from https://enterfea.com/linear-buckling-explained/”, because the Author of the website is Dr. Łukasz Skotny. In addition, no 20 and no 51 are one publication (it's the same website!)

It was modified.

The whole list of literature should be redrafted, because right now there is chaos.

It was completely revised.

In my opinion, items 20 and 28 (websites) are not sources to which Authors of scientific papers should refer, the same as the websites of companies offering commercial software for structural calculations. I am not trying to question the credibility of the content, but it cannot be the basis of scientific studies.

It was modified.

In line 168 should be “[22]” instead of “[22”.

It was modified.

Figure 2.2.2.1. needs to be amended – parts (a) and (b) should be harmonised!

It was  amended accordingly.

Figure 6.1 also contains errors – already in the first step of “design flowchart” is written “Constuct” and should be “Construct”.

It was  modified. 

The entire manuscript must undergo a thorough editorial revision.

The  manuscript was completely revised.

Reviewer 2 Report

- General comment

In my opinion, the topic of the article is interesting and could be relevant for future developments. However, it should be further developed in some aspects, such as making the text clearer, perhaps adding an example and some extra care in the figures and tables.

The article is well structured but could include some additional information in order to support some of the results/conclusions obtained. Regarding the references, there seem to be some issues, such as duplicate references, inconsistent formatting, some look incorrect to me, and some references are missing.

I made some comments to improve the quality of the article. I recommend that the authors take into account the observations and resubmit the article.

- Specified comments

Comment 01: page 1, line 12

I think the reference is missing in " Optus Stadium” and “CommBank StadiumX".

Comment 02: page 1, lines 13, 15 and 18

When you mention software or a standard, you must put the respective reference.

I think in this case it would be

Strand7 [18] [21] [33]

Australian Standard AS1170.0:2002 [16]

Australian Steel Structure code AS4100:2020 [8]

respectively.

Please verify.

Comment 03: page 2, line 75

The reference [7] is for "AS1170.2:2011" and not for "AS1170:2002".

Please check.

Comment 04: page 2, line 90

You have an extra "]" after [9][10][11].

Comment 05: page 3, table 2.1.1

When you indicate the Stadium you should put the reference of the origin of the presented data.

Note: it would be clearer if you added a photo/drawing of each of the coverages analysed.

Comment 06: pages 5, line 158

Wrong numbering, where "2.2.2.1..." should be "2.2.3.1...".

Check and correct all numbering going forward.

Comment 07: page 5, line 166

Figure 3.9 is indicated in the text, but it does not exist.

Please check.

Comment 08: page 5, line 168

The "]" is missing from the reference "[22]".

Comment 09: page 5, line 169

In the text it refers to section 3.7.2, but this section does not exist.

Please check.

Comment 10: page 5, line 194

Indicates reference [27], but it seems to me that it is the same as reference [18].

Please check and, if it is repeated, correct the situation.

Comment 11: page 6, figure 2.2.2.1

The graphs in the figure must have the same formatting.

Why not put these two graphs in one with a legend identifying the 3 curves?

Comment 12: page 7, line 236

I think the reference in " respectively (Strand7)" is missing.

Comment 13: page 7, lines 246 and 248

The "( )" is missing when referring to equations 3 and 4.

Comment 14: page 7, line 264

Where is "(Figure.3.1)" shouldn't it be "(Figure 2.3.1)"? Please check.

Comment 15: page 7, line 270

In the text it says "… and 5 trusses along the short side …". Won't it be 6 trusses? At least that's what it seems to me when analysing fig 2.3.4. Please check.

Comment 16: page 8, line 278-286

In this paragraph there should be some indication that we are now talking about the roof of Optus Stadium.

Comment 17: page 8, line 286

Indicates reference [42], but it seems to me that it is the same as reference [38].

Please check and, if it is repeated, correct the situation.

Comment 18: page 12-15, table 3.1.5-3.1.14

There is no reference to these tables throughout the text. All tables must be referenced in the text.

Still regarding these tables, on page 14 you have table 3.1.122, I think it should be 3.1.12.

Comment 19: page 19, lines 472-473 and 476

Indicates “Figure 3.1”, but according to the formatting it should be “Figure 3.1.1”.

Comment 20: page 19, figure 3.1 or 3.1.1

The caption must be after the figure.

The figure could be a little bigger in order to facilitate the analysis.

Suggestion: divide this figure into 3, one for each stadium.

Comment 21: page 20, line 490-491

Where "figures 3.2-3.4" is, it should be "figures 3.2.1-3.2.3". Please check.

Comment 22: page 28, lines 667 and 677

Where "Figure 6.1" is, it should be "Figure 4.1". Please check.

Comment 23: page 29, line 685

According to the numbering, where "7. Conclusion" is, it should be "5. Conclusion". Please check.

Comment 24: page 29, line 714-719

Throughout the article it is not possible to identify the bases to support points 1 and 2 of its conclusions. Review this situation.

Comment 25: page 30, line 733

This situation is not unique here, but if you want to present variations, such as "... critical buckling loads are reduced by up to 90% for load ..." in the conclusions or in the analysis of the results, you must indicate throughout the text the calculation of these same variations. It can be, for example, through tables where it presents the values and the respective variations.

Comment 26: pages 30 and 31, in References

It seems to me that some references are duplicated, namely [18] = [27], [38] = [42] and [20] = [51] (in the latter the link seems to be the same although the reference is presented differently). Please check.

Comment 27: page 31, reference [54]

I think the author's name should be "Silvestre, N" instead of "ilvestre, N". Check and correct.

Comment 28: pages 30 and 31, in References

It should standardize the formatting of references. See for example [11] and [15] the formatting is different.

Author Response

Reviewer’s comments

Authors’ reply

Comment 01: page 1, line 12

I think the reference is missing in " Optus Stadium” and “CommBank StadiumX".

It was added.

Comment 02: page 1, lines 13, 15 and 18

When you mention software or a standard, you must put the respective reference.

I think in this case it would be

Strand7 [18] [21] [33]

Australian Standard AS1170.0:2002 [16]

Australian Steel Structure code AS4100:2020 [8]

respectively.

Please verify.

It was addressed.

Comment 03: page 2, line 75

The reference [7] is for "AS1170.2:2011" and not for "AS1170:2002".

Please check. 

It was considered.

Comment 04: page 2, line 90

You have an extra "]" after [9][10][11].

It was considered.

Comment 05: page 3, table 2.1.1

When you indicate the Stadium you should put the reference of the origin of the presented data.

Note: it would be clearer if you added a photo/drawing of each of the coverages analysed.

The first part was considered.

It not possible to present the convergence process.

Comment 06: pages 5, line 158

Wrong numbering, where "2.2.2.1..." should be "2.2.3.1...".

Check and correct all numbering going forward.

Thanks for your comments. It was modified.

Comment 07: page 5, line 166

Figure 3.9 is indicated in the text, but it does not exist.

Please check.

Figure 3.9 was deleted.

Comment 08: page 5, line 168

The "]" is missing from the reference "[22]".

It was modified.

Comment 09: page 5, line 169

In the text it refers to section 3.7.2, but this section does not exist.

Please check.

It was considered.

Comment 10: page 5, line 194

Indicates reference [27], but it seems to me that it is the same as reference [18].

Please check and, if it is repeated, correct the situation.

They are added.

Comment 11: page 6, figure 2.2.2.1

The graphs in the figure must have the same formatting.

Why not put these two graphs in one with a legend identifying the 3 curves?

The same scale was applied.

Comment 12: page 7, line 236

I think the reference in " respectively (Strand7)" is missing.

The reference was added.

Comment 13: page 7, lines 246 and 248

The "( )" is missing when referring to equations 3 and 4.

It was added.

Comment 14: page 7, line 264

Where is "(Figure.3.1)" shouldn't it be "(Figure 2.3.1)"? Please check.

It was modified.

Comment 15: page 7, line 270

In the text it says "… and 5 trusses along the short side …". Won't it be 6 trusses? At least that's what it seems to me when analysing fig 2.3.4. Please check.

It was modified.

Comment 16: page 8, line 278-286

In this paragraph there should be some indication that we are now talking about the roof of Optus Stadium

It was added ( Optus Stadium).

Comment 17: page 8, line 286

Indicates reference [42], but it seems to me that it is the same as reference [38].

Please check and, if it is repeated, correct the situation.

It was modified.

Comment 18: page 12-15, table 3.1.5-3.1.14

There is no reference to these tables throughout the text. All tables must be referenced in the text.

Still regarding these tables, on page 14 you have table 3.1.122, I think it should be 3.1.12.

All were added in the text.

Comment 19: page 19, lines 472-473 and 476

Indicates “Figure 3.1”, but according to the formatting it should be “Figure 3.1.1”.

It was modified.

Comment 20: page 19, figure 3.1 or 3.1.1

The caption must be after the figure.

The figure could be a little bigger in order to facilitate the analysis.

Suggestion: divide this figure into 3, one for each stadium.

They were modified.

Comment 21: page 20, line 490-491

Where "figures 3.2-3.4" is, it should be "figures 3.2.1-3.2.3". Please check.

They were modified.

Comment 22: page 28, lines 667 and 677

Where "Figure 6.1" is, it should be "Figure 4.1". Please check.

Thanks, it was modified.

Comment 23: page 29, line 685

According to the numbering, where "7. Conclusion" is, it should be "5. Conclusion". Please check.

It was modified.

Comment 24: page 29, line 714-719

Throughout the article it is not possible to identify the bases to support points 1 and 2 of its conclusions. Review this situation.

The suggested points were suggested based on the overall engineering judgement. However, you r comments are sound and valid.

Comment 25: page 30, line 733

This situation is not unique here, but if you want to present variations, such as "... critical buckling loads are reduced by up to 90% for load ..." in the conclusions or in the analysis of the results, you must indicate throughout the text the calculation of these same variations. It can be, for example, through tables where it presents the values and the respective variations.

We have presented our results and outcomes based on the current simulation. However, further structural details are required to provide a more precise conclusion in this section.

Comment 26: pages 30 and 31, in References

It seems to me that some references are duplicated, namely [18] = [27], [38] = [42] and [20] = [51] (in the latter the link seems to be the same although the reference is presented differently). Please check.

They were added.

Comment 27: page 31, reference [54]

I think the author's name should be "Silvestre, N" instead of "ilvestre, N". Check and correct.

Comment 28: pages 30 and 31, in References

It should standardize the formatting of references. See for example [11] and [15] the formatting is different.

It was modified.

Reviewer 3 Report

The manuscript is about proposing a design method for Lightweight Stadium Roofing Structures Based on Advanced Analysis procedures. The mauscript is interesting.

Some very small concerns should be addressed:

1 In the model definition add a paragraph about the soil structure interaction that was considered. If not please state it clearly.

2 Please make a small paragraph concerning different regulations making small comparisons with the European and American Regulation Code (for example compare the safety factors and estimate the influence to the results)

Author Response

Reviewer’s comments

Authors’ reply

1 In the model definition add a paragraph about the soil structure interaction that was considered. If not please state it clearly.

The effect of the soil interaction was applied as boundary condition. In the current research, a particular attention was devoted to design critical steel member when subjected to a different load combination.

2 Please make a small paragraph concerning different regulations making small comparisons with the European and American Regulation Code (for example compare the safety factors and estimate the influence to the results)

The current Steel Design Australian Code AS 4100 is a combination of the EU and US codes. Thus, the overall conclusion is the same.
